# Machine Learning for Pan Evaporation Modeling in Different Agroclimatic Zones of the Slovak Republic (Macro-Regions)

Beáta Novotná [1,*] , Ľuboš Jurík [1] , Ján Čimo [1] , Jozef Palkovič [2] , Branislav Chvíla [3] and Vladimír Kišš [4]

1   Institute of the Landscape Engineering, Faculty of Horticulture and Landscape Engineering,
    Slovak University of Agriculture, 949 76 Nitra, Slovakia; Lubos.Jurik@uniag.sk (Ľ.J.);
    Jan.Cimo@uniag.sk (J.Č.)
2   Institute of Statistics, Operation Research and Mathematics, Faculty of Economics and Management,
    Slovak University of Agriculture, 949 76 Nitra, Slovakia; Jozef.Palkovic@uniag.sk
3   Meteorological and Climatological Monitoring, Network of Ground Synoptic Stations,
    Slovak Hydrometeorological Institute, 833 15 Bratislava, Slovakia; branislav.chvila@shmu.sk
4   AgroBioTech Research Centre, Slovak University of Agriculture, 949 76 Nitra, Slovakia;
    vladimir.kiss@uniag.sk
*   Correspondence: Beata.Novotna@uniag.sk

**Abstract:** Global climate change is likely to influence evapotranspiration (ET); as a result, many ET calculation methods may not give accurate results under different climatic conditions. The main objective of this study is to verify the suitability of machine learning (ML) models as calculation methods for pan evaporation modeling on the macro-regional scale. The most significant PE changes in the different agroclimatic zones of the Slovak Republic were compared, and their considerable impacts were analyzed. On the basis of the agroclimatic zones, 35 meteorological stations distributed across Slovakia were classified into six macro-regions. For each of the meteorological stations, 11 variables were applied during the vegetation period in the years from 2010 to 2020 with a daily time step. The performance of eight different ML models—the neural network (NN) model, the autoneural network (AN) model, the decision tree (DT) model, the Dmine regression (DR) model, the DM neural network (DM NN) model, the gradient boosting (GB) model, the least angle regression (LARS) model, and the ensemble model (EM)—was employed to predict PE. It was found that the different models had diverse prediction accuracies in various geographical locations. In this study, the results of the values predicted by the individual models are compared.

**Keywords:** pan evaporation; agroclimatic zone; macro-region; climatic characteristic; machine learning

## 1. Introduction

Changes in climatic factors, such as in temperature, wind speed, sunshine hours, humidity, and solar radiation, can have a significant impact on the evapotranspiration (ET) process [1]. ET is the combination of two separate water loss processes: water's evaporation from the soil and plant surfaces and plant transpiration, through which water escapes from a plant's body into the ambient air through its stomata in the form of steam [2,3]. Evaporation is an important component of the hydrological cycle [4]. Pan evaporation (PE) is extensively used in developing irrigation projects and provincial water resources [5,6]. PE is an important climatic variable for developing efficient water resource management strategies [7]. Generally, two approaches—(i) direct (i.e., pan evaporimeter) and (ii) indirect (i.e., empirical or semi-empirical equations)—are used to measure evaporation [8,9]. The direct estimation of PE using a class A pan evaporimeter has limited spatial coverage because of practical and instrumental problems [9–11]. In contrast, the application of the indirect PE estimation method based on the relationship of ET with various climatic parameters is often restricted due to data availability and climate variability [9,12,13]. Considering the limitations of both the methods, machine learning (ML) techniques have

been used in recent years as an alternative [9]. In the past, many ML models for PE modeling using different combinations of available climatic variables have been reported in the literature [13]. Owing to the rapid development of ML techniques, a series of powerful tools has been proposed over the last two decades, allowing the scientific community to obtain new insights into the patterns of ET on different spatial scales, ranging from ecosystem to global. Elucidating the biophysical mechanisms governing the exchange of water vapor between land and the atmosphere is particularly crucial for addressing water scarcity under climate change [14]. Some applications of ML models for PE estimation are described in [1,9,13,15,16]. In the last decade, advances in computation have led to the introduction of ML methodologies for referencing evapotranspiration calculations, and the high accuracy of their results has been proven by using different approaches [17]. In the last few decades, ML techniques have been increasingly utilized to estimate hydrological variables [18–22], ecological variables [23], and renewable energy variables [24], as described in [14]. Because ML techniques solve the non-linear relationships between input and output variables [14], many ML techniques have been proposed to estimate ET for hydrological applications [25], such as k-nearest neighbors [26], support vector machines [27], random forests [26], and artificial neural networks (ANNs) [28]. Previously, most studies applied ML approaches to in situ measurements; however, many recent studies have also applied ML approaches to remote sensing data [29–31], as shown in [32], as well as [33,34].

Potential evapotranspiration represents the upper limit of ET when this process is not limited by water deficit in the soil [35]. Potential evapotranspiration in Slovakia is estimated according to empirical or semi-empirical relationships based on measurements of other meteorological elements, as described by the authors of [36], who compared the daily reference crop's (grass cover) potential evapotranspiration results which were calculated with two modifications of the Penman–Monteith equation. The authors of [37] studied drought occurrence using the Standardized Precipitation and Evapotranspiration Index (SPEI) and the Standardized Precipitation Index (SPI). Maps were constructed based on data calculated using the Budyko–Zubenokova method in 31 Slovak climatological stations [35].

Several local-scale studies have been conducted in the Slovak Republic, e.g., [35,38,39]; however, an overall PE estimation with ML techniques and by using all of the available pan vapor data for the whole territory of the country is missing, and no similar works have been carried out by using these kinds of methods. In order to obtain an overview about the present state of research in this field, a review of related research articles was carried out, and details of some recently published articles are listed in Table 1.

**Table 1.** Example of ML applications for PE estimation.

| Author | Country | Recommended Model for PE Estimation | Input Climatic Data Variables * | Statistical Indices ** | Number of the Meteorological Stations |
|---|---|---|---|---|---|
| Majhi and Naidu (2021) [13] | India | functional link artificial neural network (FLANN) | PE, $T_{max}$, $T_{min}$, RH1, RH11 | RMSE = 0.85; MAE = 0.63; EF = 0.70 | 3 |
| Kisi, O. (2015) [40] | Mediterranean Region of Turkey | multivariate adaptive regression splines (MARS), M5 Model Tree (M5Tree) | PE, $T_{aver}$ SR, RH, Us | RMSE = 0.189 | 2 |
| Zounemat-Kermani et al. (2021) [16] | Turkey | Levenberg–Marquardt (MLP-LM) | PE, $T_{max}$, $T_{min}$, SR, S, RH, Us | MAE = 0.492; d = 0.981 | 2 |
| Malik et al. (2021) [9] | Northern India | Slap Swarm Algorithm (SVR-SSA) | PE, $T_{max}$, $T_{min}$, $RH_{max}$, Rhmin, SR, Us | MAE = 0.697; RMSE = 1.1; IOS = 0.250; NSE = 0.861; PCC = 0.929; IOA = 0.960 | 3 |

**Table 1.** *Cont.*

| Author | Country | Recommended Model for PE Estimation | Input Climatic Data Variables * | Statistical Indices ** | Number of the Meteorological Stations |
|---|---|---|---|---|---|
| Abed et al. (2021) [41] | Malaysia | Long Short-Term Memory Neural Network (LSTM) | PE, $T_{aver}$, $T_{max}$, $T_{min}$, RH, SR, Us | $R^2$ = 0.970; MAE = 0.135; MSE = 0.027; RMSE = 0.166; RAE = 0.173; RSE = 0.029 | 2 |
| Ferreira et al. (2019) [42] | Brazil | multivariate adaptive regression splines (MARS) | Etr, $T_{aver}$, SR, Us, G, es, ea, $\Delta$, y | $R^2$ (0.79–0.85); RMSE (0.41–0.54); MAE (0.34–0.46) | 8 |
| Al-Mukhtar (2021) [1] | middle, south, and north of Iraq | weighted K-nearest neighbor (KKNN) | PE, $T_{max}$, $T_{min}$, T, RH, Us | $R^2$ = 0.98; RMSE = 26.39; MAE = 18.62; NSE = 0.97; PBIAS = 3.8 | 3 |
| Wang et al. (2017) [11] | China | multiple linear regression (MLR), Stephens and Stewart model (SS) | PE, $T_{aver}$, SR, S, RH, Us | $R^2$ = 0.988; RSME (0.314–0.405) | 8 |
| Sattari et al. (2021) [43] | Northwest Iran | M5 tree model (M5Tree) | PE, $T_{aver}$, RH, Us, P | RMSE (0.0042–0.0058); $R^2$ (0.9916–0.9952); $t$-test (0.722–0.96); NSE (0.989 to 0.994) | 4 |
| Adnan et al. (2017) [44] | Pakistan | principal component analysis (PCA) | PE, $T_{max}$, $T_{min}$, $T_{aver}$ RH, SR, Us, P | R = 0.83426 | 1 |
| Simon-Gáspár et al. (2021) [45] | Hungary | multiple stepwise regression (MLR) | PE, $T_{aver}$, $T_{max}$, $T_{min}$, RH, Us, Rs | RMSE = 0.834; MAE = 0.660; S = 0.217 | 1 |

* Pan evaporation (PE); reference evapotranspiration (Etr); average, maximum, minimum, morning, and afternoon relative humidity (RH, $Rh_{max}$, $Rh_{min}$, RH1, and RH11); average, maximum, and minimum air temperature ($T_{aver}$, $T_{min}$, and $T_{max}$); wind speed (Us); relative sunshine duration (S); solar radiation (SR); soil heat flux (G); saturation vapor pressure (es); actual vapor pressure (ea); slope of the saturation vapor pressure function ($\Delta$); psychometric constant (y); precipitation (P); global radiation (Rs). ** Root mean square error (RMSE); mean absolute error (MAE); efficiency factor (EF); Willmott index (d); absolute error (MSE); determination coefficient (R2); regression value (R); relative absolute error (RAE); relative squared error (RSE); percentage bias (PBias); unpaired two-sample $t$-test ($t$-test); Nash–Sutcliffe efficiency index (NSE); scatter index (SI).

The North Atlantic Stream, as part of the Gulf Stream, has a favorable effect on the climatic conditions of Western and Northwestern Europe. These effects are mainly reflected in the reduction in temperature fluctuations throughout the year; this means that, for example, during the winter period, the air temperatures do not decrease below the freezing point [46]. However, the weather in Slovakia also affects the flows of air from the Atlantic Ocean and from the Adriatic Sea. The individual weather forecasting models used in Slovakia have also contradicted each other in certain time periods. An example is the year 2021, when weather differed from most of the years of the second decade of the 20th century. Frequent changes in weather were typical for most of the year 2021. This has occurred since the end of summer and during autumn, and it was also characteristic for the month of December. At the beginning of this month, it warmed up, then there was a cooling period; however, this only occurred in some parts of Slovakia, for example in the west and southwest, and it was accompanied by heavy snowfall. In the middle of the month of December, it warmed noticeably, and it lasted until the end of the year [47]. The climate of a particular area is also affected by microclimatic factors, especially the shape of the relief (convex or concave), the orientation of the relief towards the world and the prevailing flow, relative altitude fragmentation, vegetation, and anthropogenic influences. The aim of this study is to obtain information about the areas that are the most affected by PE changes, which is closely related to the manifestations of global climate change.

For the purpose of this study, eight newly explored ML models for estimating PE losses in six macro-regional scales were assessed. The macro-regions were classified according to individual agroclimatic zones. Using available climatic data from 35 different meteorological stations across the Slovak Republic, we performed the estimation of the PE in the vegetation period applying the datasets selected for the years 2010 to 2020, including their different elevations above sea level and geographical locations.

As a result of the above-mentioned literature resources, ML is an important tool for the goal of leveraging technologies around artificial intelligence. ML is, in part, based on a model of brain cell interaction. Although the model was created in 1949 by Donald Hebb (1949) [48] in a book titled *The Organization of Behavior* (Keith, 2021) [49], the applications of ML techniques to evapotranspiration estimation problems are currently limited and the knowledge on the topic is still partial and fragmented [50].

The results of this study are unique because: (1) there is no comparative study for the whole of the Slovak Republic, where all available climate data for the last climate decade for PE estimation are applied; (2) the new progressive ML method was used for PE assessment; and (3) besides the selection of the applicability and predictability of eight different ML models, the macro-regions with the most pronounced manifestations of global climate change are identified.

## 2. Materials and Methods

### 2.1. Study Location and Climatic Data Collection

In connection with global climate change, the gradual modification of the energy balance globally, and especially locally, was observed. Therefore, the assessment of PE changes and their trends in the Slovak Republic has its justification. The climatic data from the climatological network of Slovak stations were provided by the Slovak Hydrometeorological Institute (SHMI) in Bratislava. The available PE dataset for the vegetation periods of the years from 2010 to 2020 was analyzed according to individual regions of the Slovak Republic.

The daily data of PE during the main vegetation period (from April to October) from 35 meteorological stations distributed across Slovakia (see Figure 1) were used in this study. There were evaluated data mostly from the ten consecutive years (2010–2020). All available PE data measured in the Slovak Republic by SHMI during the observed periods were applied in this study. The reason for this is not only the availability of the data, but also the application of the same instrument for their measurement, i.e., a GGI—3000 evaporator (see principle in [51]).

Slovak Republic Zoning Criteria

The zoning criteria are based on the agroclimatic division of Slovakia [52]. For the agroclimatic division of Slovakia, three basic agroclimatic indicators were used, according to the area divided into agroclimatic macro-areas, sub-areas, and districts.

(a) The agroclimatic temperature indicator (TS10) is the sum of average daily air temperatures during the period with an average daily temperature of $\geq 10.0\ ^\circ$C. According to this indicator, the territory of Slovakia is divided into three agroclimatic macro-areas and eight agroclimatic areas:

1. Warm agroclimatic macro-area with TS10 from 3100 to 2400 $^\circ$C;
2. Slightly warm agroclimatic macro-region with TS10 from 2400 to 2000 $^\circ$C; and
3. Cold agroclimatic macro-region with TS10 from 2000 to 1600 $^\circ$C.

(b) The agroclimatic moisture indicator ($K_{VI–VIII}$) is given by the difference of potential evaporation (Eo) and precipitation (P) in the summer months (June–August (VI–VIII)):

$$K_{VI–VIII} = E_o − P \text{ (mm)} \tag{1}$$

$E_o$ values were calculated for the territory of Slovakia by [53]. The climatic indicator "K" expresses the moisture balance of the area well (expressed in millimeters). Positive

values $E_o − P$ are characterized by the water deficiency and negative values by the moisture excess.

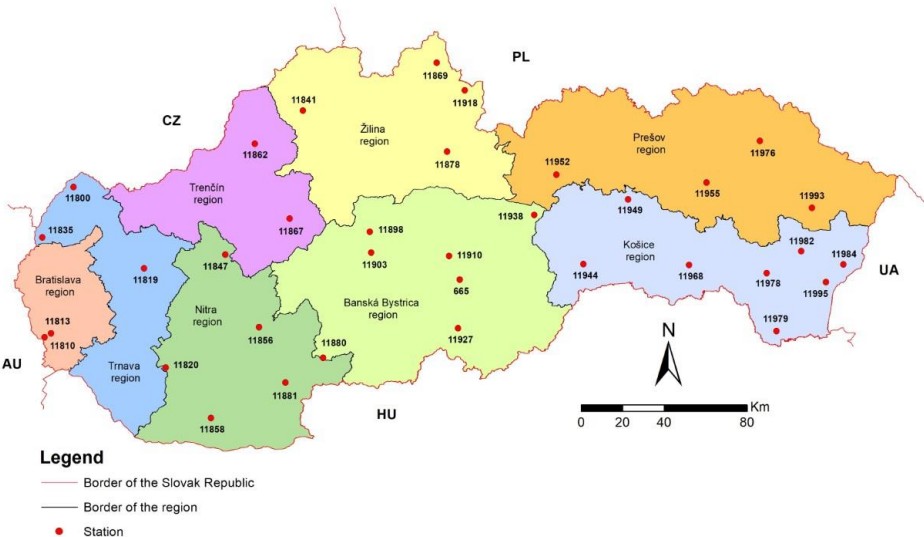

**Figure 1.** Geographical location of the used climatological stations in the Slovak Republic divided into eight regions and their macro-regional classification: northwest (NW): Trenčín region; southwest (SW): Trnava region, Bratislava region, Nitra region; north-central (NC): Žilina region; south-central (SC): Banská Bystrica region; northeast (NE): Prešov region; southeastern (SE): Košice region.

According to the climatic indicator ($K_{VI–VIII}$), there are seven sub-areas in Slovakia, in which agricultural crops have differently ensured their moisture requirements:

1.   Subarea with $K_{VI–VIII} \geq 150$ mm—very dry
2.   Subarea with $K_{VI–VIII}$ 150 to 100 mm—mostly dry
3.   Subarea with $K_{VI–VIII}$ 100 to 50 mm—slightly dry
4.   Subarea with $K_{VI–VIII}$ 50 to 0 mm—slightly humid
5.   Subarea with $K_{VI–VIII}$ 0 to −50 mm—mostly humid
6.   Subarea with $K_{VI–VIII}$ −50 to −100 mm—humid
7.   Subarea with $K_{VI–VIII}$ −100 mm—very humid

(c) The agroclimatic wintering indicator ($T_{min}$) represents the average of the annual absolute temperature minima. This characteristic effectively describes the climatic conditions during the winter. Absolute temperature minima are important factors for the cultivation of winter crops and fruit trees. They effectively express the critical freezing temperatures. According to the conditions of wintering, there are five districts:

1.   Agroclimatic district of mostly mild winter with $T_{min} \geq −18$ °C;
2.   Agroclimatic district of relatively mild winters with $T_{min}$ from −18.0 °C to −20.0 °C;
3.   Agroclimatic district of mildly cold winter with $T_{min}$ from −20.0 °C to −22.0 °C;
4.   Agroclimatic district of mostly cold winter with $T_{min}$ from −22.0 to −24.0 °C;
5.   Agroclimatic district of cold winter with $T_{min} \leq −24.0$ °C.

The average annual vertical temperature gradient in Slovakia is 0.61 °C per 100 m of height. During the summer months, however, its value increases to 0.76 °C and, in the winter months, it decreases to 0.33 °C. The territory of the Slovak Republic was divided into eight regions (see Figure 1). The classification of the eight Slovak regions into six macro-regions with 35 individual meteorological stations across Slovakia is shown in Table 2.

According to Figure 1 and Table 2, the division of eight regions into six macro-regions according to the different agroclimatic zones is as follows:

● Northwest (NW): Trenčín region;
● Southwest (SW): Trnava region, Bratislava region, and Nitra region;

- North-central (NC): Žilina region;
- South-central (SC): Banská Bystrica region;
- Northeast (NE): Prešov region;
- Southeastern (SE): Košice region.

The characteristics of the agroclimatic zones were applied to classify eight regions into six macro-regions, which are:

(1) Northwestern Slovakia (NW): Trenčín region—From an agroclimatic point of view, the NW area is assigned to the macro-area of a mildly warm, agroclimatic area of a lightly warm subarea that is moderately humid to mostly humid. Moreover, this area is assigned to agroclimatic precincts with a mild/cold winter to a mostly cold winter. This area also transitions north to the macro-area with a cold, agroclimatic area with a moderately cold sub-area that is mostly humid to humid, and a precinct with mostly cold winters.

(2) Southwestern Slovakia (SW): Bratislava region, Trnava region, and Nitra region— From an agroclimatic point of view, the SW area is assigned to the macro-area of a warm and agroclimatic area that is very warm, a sub-area that is very dry, and a predominantly dry and agroclimatic precinct that has mainly mild winters.

(3) North-central Slovakia (NC): Žilina region—From an agroclimatic point of view, the NC area is assigned to the macro-area of a slightly warm to cold agroclimatic area from a slightly to moderate/mildly warm sub-area up to a slightly cold sub-area. This area is also assigned to slightly dry, moderately humid, mostly humid, and agroclimatic precincts that are mildly cold to mostly cold in the winter. This area transitions north to a macro-area of a cold, agroclimatic area that is mostly cold, a sub-area that is mostly humid to humid, and a precinct that is mostly cold/cold in the winter.

(4) South-central Slovakia (SC): Banská Bystrica region—From an agroclimatic point of view, the SC area at the southernmost part of the state border is assigned to a warm macro-area, a very warm agroclimatic area, a very dry and predominantly dry sub-area, and an agroclimatic precinct with a predominantly mild winter. This area transitions to a macro-area with a moderately warm, an agroclimatic area that is relatively mild/warm, a sub-area that is slightly humid to mostly humid, and a precinct that is slightly cold in the winter.

(5) Northeastern Slovakia (NE): Prešov region—From an agroclimatic point of view, this area is the most diverse. In the southern part, it is considered a warm macro-area, with an agroclimatic area that is sufficiently warm to relatively/moderately warm, sub-areas that are predominantly dry to moderately dry, and agroclimatic precincts that have relatively mild winters to mild cold winters. This area transitions north to a macro-area that is warm or moderately warm to cold, an agroclimatic area that is relatively mild/warm to slightly cold, sub-areas that are moderately dry or slightly humid, and a precinct that is slightly cold in the winter to mostly cold in the winter.

(6) Southeastern Slovakia (SE): Košice region—From an agroclimatic point of view, the SE area is assigned as the macro-area of a warm and agroclimatic area that is very warm, a sub-area that is very dry and a predominantly dry, and a agroclimatic precinct with a predominantly mild winter. This area transitions to an agroclimatic area that is mostly warm, a sub-area that is very dry, and a precinct that has a relatively mild winter.

Table 2. Identification of the individual regions across the territory of the Slovak Republic and their integration according to macro-regions.

| Number | Identification | Station Name | Pan Evaporation Measurement | | Classification | Region Identification | Classification | Region Identification | Macro-region Classification | | |
|---|---|---|---|---|---|---|---|---|---|---|---|
| | | | Setting date | Ending date | | | | | | | |
| 1 | 665 | ĎUBÁKOVO | 1 May 2011 | 30 June 2016 | BB (Banskobystrický) | SC (south of central Slovakia) | NW (north of western Slovakia) | TN (Trenčiansky) | NW | NC | NE |
| 2 | 11800 | HOLÍČ | 1 April 2011 | 30 June 2015 | TT (Trnavský) | SW (south of western Slovakia) | SW (south of western Slovakia) | TT (Trnavský) | SW | SC | SE |
| 3 | 11810 | BRATISLAVA–M. DOLINA | 1 April 2011 | 31 October 2020 | BA (Bratislavský) | SW (south of western Slovakia) | | BA (Bratislavský) | | | |
| 4 | 11813 | BRATISLAVA-KOLIBA | 1 April 2011 | 31 October 2011 | BA (Bratislavský) | SW (south of western Slovakia) | | NR (Nitriansky) | | | |
| 5 | 11819 | JASLOVSKÉ BOHUNICE | 1 April 2011 | 31 July 2011 | TT (Trnavský) | SW (south of western Slovakia) | NC (north of central Slovakia) | ZA (Žilinský) | | | |
| 6 | 11820 | ŽIHÁREC | 1 April 2011 | 31 October 2020 | NR (Nitriansky) | SW (south of western Slovakia) | SC (south of central Slovakia) | BB (Banskobystrický) | | | |
| 7 | 11835 | MORAVSKÝ SVÄTÝ JÁN | 1 April 2011 | 31 May 2016 | TT (Trnavský) | SW (south of western Slovakia) | NE (north of eastern Slovakia) | PO (Prešovský) | | | |
| 8 | 11841 | DOLNÝ HRIČOV | 1 April 2011 | 30 September 2011 | ZA (Žilinský) | NC (north of central Slovakia) | SE (south of eastern Slovakia) | KE (Košický) | | | |
| 9 | 11847 | TOPOĽČANY | 1 April 2011 | 31 October 2020 | NR (Nitriansky) | SW (south of western Slovakia) | | | | | |
| 10 | 11856 | MOCHOVCE | 1 April 2011 | 31 July 2011 | NR (Nitriansky) | SW (south of western Slovakia) | | | | | |
| 11 | 11858 | HURBANOVO | 1 April 2011 | 31 October 2020 | NR (Nitriansky) | SW (south of western Slovakia) | | | | | |

**Table 2.** *Cont.*

| Number | Identification | Station Name | Pan Evaporation Measurement | | Classification | Region Identification | Classification | Region Identification | Macro-region Classification |
|--------|----------------|--------------|------|------|----------------|----------------------|----------------|----------------------|------------------------------|
| 12 | 11862 | BELUŠA | 1 April 2011 | 31 August 2015 | TN (Trenčiansky) | NW (north of western Slovakia) | | | |
| 13 | 11867 | PRIEVIDZA | 1 April 2011 | 31 October 2020 | TN (Trenčiansky) | NW (north of western Slovakia) | | | |
| 14 | 11869 | RABČA | 1 May 2011 | 31 October 2020 | ZA (Žilinský) | NC (north of central Slovakia) | | | |
| 15 | 11878 | LIPTOVSKÝ MIKULÁŠ | 1 May 2011 | 31 October 2020 | ZA (Žilinský) | NC (north of central Slovakia) | | | |
| 16 | 11880 | DUDINCE | 1 April 2011 | 31 October 2020 | BB (Banskobystrický) | SC (south of central Slovakia) | | | |
| 17 | 11881 | ŽELIEZOVCE | 1 June 2011 | 31 October 2014 | NR (Nitriansky) | SW (south of western Slovakia) | | | |
| 18 | 11898 | BANSKÁ BYSTRICA | 1 May 2011 | 31 October 2020 | BB (Banskobystrický) | SC (south of central Slovakia) | | | |
| 19 | 11903 | SLIAČ | 1 April 2011 | 31 October 2020 | BB (Banskobystrický) | SC (south of central Slovakia) | | | |
| 20 | 11910 | LOM NAD RIMAVICOU | 1 May 2011 | 31 October 2020 | BB (Banskobystrický) | SC (south of central Slovakia) | | | |
| 21 | 11918 | LIESEK | 1 April 2011 | 31 October 2020 | ZA (Žilinský) | NC (north of central Slovakia) | | | |
| 22 | 11927 | BOĽKOVCE | 1 April 2011 | 31 October 2020 | BB (Banskobystrický) | SC (south of central Slovakia) | | | |
| 23 | 11938 | TELGÁRT | 1 May 2011 | 12 October 2020 | BB (Banskobystrický) | SC (south of central Slovakia) | | | |
| 24 | 11944 | ROŽŇAVA | 1 April 2011 | 30 October 2017 | KE (Košický) | SE (south of eastern Slovakia) | | | |
| 25 | 11949 | SPIŠSKÉ VLACHY | 1 May 2011 | 31 October 2020 | KE (Košický) | SE (south of eastern Slovakia) | | | |
| 26 | 11952 | GÁNOVCE | 20 April 2011 | 30 October 2020 | PO (Prešovský) | NE (north of eastern Slovakia) | | | |
| 27 | 11955 | PREŠOV-VOJSKO | 1 May 2011 | 31 October 2020 | PO (Prešovský) | NE (north of eastern Slovakia) | | | |

**Table 2.** *Cont.*

| Number | Identification | Station Name | Pan Evaporation Measurement | | Classification | Region Identification | Classification | Region Identification | Macro-region Classification |
|---|---|---|---|---|---|---|---|---|---|
| 28 | 11968 | KOŠICE, LETISKO | 1 April 2011 | 31 October 2020 | KE (Košický) | SE (south of eastern Slovakia) | | | |
| 29 | 11976 | TISINEC | 1 April 2011 | 31 October 2020 | PO (Prešovský) | NE (north of eastern Slovakia) | | | |
| 30 | 11978 | TREBIŠOV, MILHOSTOV | 1 April 2011 | 30 October 2020 | KE (Košický) | SE (south of eastern Slovakia) | | | |
| 31 | 11979 | SOMOTOR | 1 April 2011 | 31 October 2014 | KE (Košický) | SE (south of eastern Slovakia) | | | |
| 32 | 11982 | MICHALOVCE | 1 April 2011 | 31 October 2020 | KE (Košický) | SE (south of eastern Slovakia) | | | |
| 33 | 11984 | ORECHOVÁ | 1 April 2011 | 31 October 2020 | KE (Košický) | SE (south of eastern Slovakia) | | | |
| 34 | 11993 | KAMENICA N. CIROCHOU | 1 April 2011 | 31 October 2020 | PO (Prešovský) | NE (north of eastern Slovakia) | | | |
| 35 | 11995 | VYSOKÁ NAD UHOM | 1 April 2011 | 31 October 2012 | KE (Košický) | SE (south of eastern Slovakia) | | | |

Extreme temperatures are considered a limitation for the production process of field and garden crops and often for all agricultural production. The absolute maximum temperature in Slovakia was measured on 20 July 2017 in Hurbanovo at 40.3 °C and the absolute minimum temperature was measured on 11 February 1929 in Vígľaš-Pstruša at −41.0 °C.

## 2.2. Case Study and Data Description

In this study, the following measured meteorological data were applied on a daily basis: (1) daily pan evaporation in millimeters (PE); (2) minimum temperature in degrees Celsius ($T_{min}$); (3) maximum temperature in degrees Celsius ($T_{max}$); (4) average temperature in degrees Celsius ($T_{aver}$); (5) relative humidity as a percentage (RH); (6) average wind speed in meters per second (Us) with (7) wind speed measured at 7, 14, and 21 h, as well as (8) prevailing wind directions measured at 7, 14, and 21 h; (9) the sum of precipitation in millimeters (P); and (10) vapor pressure in hectopascals (E). For each station, the (11) elevation above sea level in meters (elevation) and (12) geographical location (coordinates) were applied.

## 2.3. ML Models and Evaluation Criteria

The data collected daily from 35 meteorological stations were processed into an extensive database (including about 176,000 observation records) in the form of panel data. Each observation was labelled by time and cross-sectional characteristics. The database was divided into a ratio of 40:30:30 into data for model estimation (train), for model validation (Validate) and for model testing (Test). The analysis was performed in SAS Enterprise Miner 15.1. As part of the workflow, the most used modeling techniques for large-scale data analysis were applied. As the target variable, pan evaporation was selected. All the other variables were used as the input variables, including geographical location. Based on the input data, the following models were estimated, validated, and tested.

### 2.3.1. Neural Network (NN)

The neural network with multilayer perception (MLP) model architecture was tested. Artificial neurons are similar to biological neurons in terms of learning based on experience and generalizations of previous experiences and applying acquired experience to new data. MLP is a neural network inspired by the structure of the real brain [54]. It consists of layers that contain neurons as processing units. Each unit in a layer is associated with all units in the previous layer. The knowledge of the whole network is encoded in the weights of these connections. Such a neural network can accurately predict the outputs of any unknown function. The network uses three nodes (input layer, hidden layer, and output layer), which are neurons with a nonlinear activation function [55] (see Figure 2).

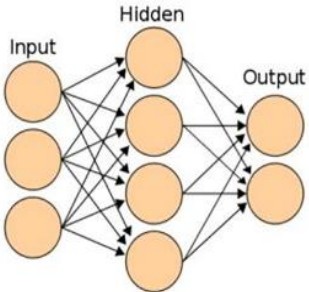

**Figure 2.** Architecture of a convolutional neural network according to [56].

### 2.3.2. AutoNeural Network (AN)

An autoneural network is a modeling tool in which the neural network architecture is automatically configured by the software to achieve the best result. The main advantage of a neural network is its ability to detect complex nonlinear relationship between dependent

and independent variables and all possible interactions between them [57]. Another advantage of neural networks is the availability of multiple training algorithms which may be applied without deeper statistical knowledge. As disadvantages could be considered a higher computational burden, it has the empirical nature of model development and the tendency towards overfitting. Another disadvantage is the "black box" nature of neural networks, which complicates the further modification and deeper interpretation of model results compared to other types of models.

### 2.3.3. Decision Tree (DT)

An empirical decision tree represents a segmentation of data that is created based on a series of simple decision rules. The tree is created based on an algorithm that identifies different ways of dividing the dataset into individual segments similar to branches. They can be used for the analysis of categorical as well as quantitative variables. The rule for forming the branches of the decision tree is based on the method of extracting the relationship between the analysis object and one or more inputs to create the segments [58].

### 2.3.4. Dmine Regression (DR)

Dmine regression is an estimation method using a sequentially applied least squares method. In each step, an independent variable is selected that contributes most to the explained model's variability. In the estimation, bidirectional interactions between classification variables are considered, a possible non-linear relationship between the input variables and the target variable is identified, and group variables are used to reduce the number of the classification variable levels.

### 2.3.5. DM Neural Network (DM NN)

A DM neural network is the method used to estimate the additive nonlinear model, whereby the main components are used to predict the target variable. In contrast to the neural network, it overcomes the problem of nonlinear estimation and the identification of the global optimal solution with a significantly shorter computational time. In the first step, the method of the main components is applied to the training data, from which a smaller number of components is subsequently selected for use in the next modeling process. At each stage of the modeling process, eight training functions are estimated separately in the training dataset, from which the variant that achieves the best results is selected.

### 2.3.6. Gradient Boosting (GB)

Gradient boosting is the application of the algorithm published by [59]. The algorithm searches for the optimal partition of data defined in terms of the values of one variable. The optimality of the criterion depends on how the target variable is divided into the individually defined partitions. The more similar the values of the target variable within a segment, the higher the value of the partition. The majority of the similar algorithms work through a process called recursive partitioning. The partitions are then combined to create a predictive model. The accuracy of the model is evaluated by standard indicators based on a comparison with the values of the target variable. As with decision trees, there is no assumption about the distribution of variables that need to comply with the gradient boosting. For interval input variables, the model only works with variable sequences. For analyzed data, where decision trees have achieved good results, gradient boosting can often further improve this result.

### 2.3.7. Least Angle Regression (LARS)

This type of regression was first published by Efron et al. (2004) [60]. Similar to the forward selection that is often used in regression models, the algorithm produces a sequence of regression models. One additional parameter is added at each step. The sequential procedure is completed after all parameters have entered the model. The algorithm begins by centering the input and output variables. The input variable scales are adjusted to the

same number of corrected squares. The initial level of the coefficients is zero, as is the predicted effect on the output variable. Subsequently, the predictor that is most correlated with the residues is identified and a step in the direction of this predictor follows. The step length determines the predictor coefficient. The step length is determined so that any of the other predictors and the current predicted effect have the same correlation with the residues. At this point, the predicted effect moves in a direction that is rectangular between the two predictors. This will ensure that already mentioned predictors have the same correlation with residues. The predicted effect moves in the same direction until the third predictor also has the same correlation with residues as the previous two predictors previously in the model. The new direction is designed to be rectangular between the three predictors. The predicted effect moves again in this direction until a fourth predictor that has the same correlation with residues enters the model. In this way, the process continues until all predictors enter the model.

2.3.8. Least Ensemble Model (EM)

This type of model is one of the methods that combines posterior probabilities and two or more predictive models to create a potentially more accurate model [61]. In the case of the analysis performed, the ensemble model was based on the results of all the above models.

The modeling process is shown in Figure 3. In the initial step, data were collected and processed into the input database. The database was divided into train, validate, and test datasets. In the next step, all the modeling techniques described above in the methodology were applied in the train dataset. Estimated models were validated and tested using validate and test datasets. In the final step, the best model was selected based on the smallest value of average squared error achieved in validation and testing.

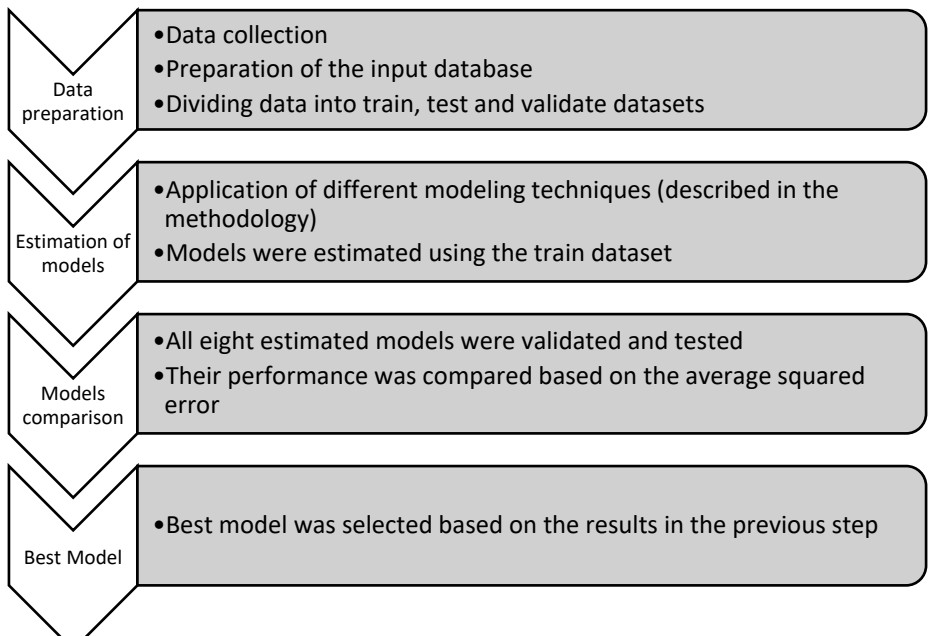

**Figure 3.** Workflow of the modeling process.

## 3. Results

### 3.1. PE Changes at the Macro-Regional Level

Table 3 shows the basic characteristics of PE for individual regions. On average, the lowest PE values were recorded in the NC area, and the highest were recorded in the SE and SW regions. As for the largest change over the period, it could be expressed using the variation range in the last column. It follows that the largest change in terms of PE occurred

in the SE region, followed by SC and SW. The smallest change in terms of evaporation was in the NC and NW regions. Similar conclusions can be taken from Figure 4.

**Table 3.** Overview of the individually analyzed regions of the Slovak Republic with the statistic assessment of their evaluated elements.

| Analysis Variable: Pan Evaporation (PE) (mm) | | | | | | |
|---|---|---|---|---|---|---|
| Region Orientation | Observation Numbers | Mean | Median | Standard Deviation | Coefficient of Variation | Range |
| SW | 23,723 | 2.55 | 2.40 | 1.43 | 56.27 | 14.00 |
| SC | 18,557 | 2.28 | 2.20 | 1.20 | 52.76 | 15.20 |
| NC | 11,128 | 1.96 | 1.90 | 1.10 | 56.18 | 7.00 |
| SE | 22,256 | 2.55 | 2.40 | 1.51 | 59.35 | 18.50 |
| NE | 11,128 | 2.32 | 2.20 | 1.23 | 53.02 | 12.60 |
| NW | 5381 | 2.30 | 2.20 | 1.37 | 59.60 | 8.50 |

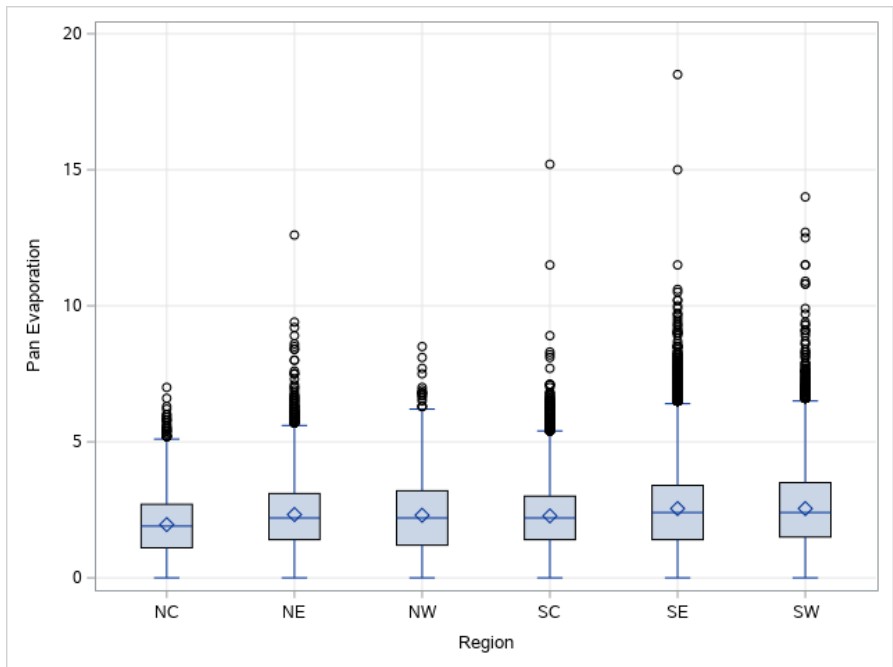

**Figure 4.** This figure shows the comparison of the individual analyzed regions of the Slovak Republic with the minimum and maximum values of the PE (mm). The box is bounded by the lower and upper quartile, the line in the middle is the median, and the diamond means the average value.

Figure 4 compares the measured PE values in the individual regions. For each region, Figure 4 shows the minimum and maximum values, the box is bounded by the lower and upper quartile, the line in the middle is the median, and the diamond means the average value. Values that are displayed as points above the graph are considered extreme values. The values are in accordance with Table 4, where the lowest value on average is in NC, and the highest values are in SW and SE. The displayed values represent the measurements taken during the vegetation season. In terms of the graphical range of values, similar results were also achieved, either with or without the consideration of extreme values. The highest margin is in SE and SW, or SC, where the highest PE changes were found. If we assess the differences from a statistical point of view, Table 4 shows the differences between areas that can be considered statistically significant.

**Table 4.** Overview of the statistically significant differences in PE according to six macro-regions: northwest (NW), southwest (SW), north-central (NC), south-central (SC), northeast (NE), and southeast (SE).

| Region Comparison | Difference between Means | Simultaneous 95% Confidence Limits | | |
|:---:|:---:|:---:|:---:|:---:|
| SE—SW | 0.00009 | −0.04029 | 0.04047 | |
| SE—NE | 0.22104 | 0.17463 | 0.26744 | *** |
| SE—NW | 0.24533 | 0.18232 | 0.30834 | *** |
| SE—SC | 0.26869 | 0.22804 | 0.30933 | *** |
| SE—NC | 0.58997 | 0.54258 | 0.63735 | *** |
| SW—SE | −0.00009 | −0.04047 | 0.04029 | |
| SW—NE | 0.22094 | 0.17405 | 0.26783 | *** |
| SW—NW | 0.24523 | 0.18186 | 0.30860 | *** |
| SW—SC | 0.26859 | 0.22739 | 0.30980 | *** |
| SW—NC | 0.58987 | 0.54201 | 0.63773 | *** |
| NE—SE | −0.22104 | −0.26744 | −0.17463 | *** |
| NE—SW | −0.22094 | −0.26783 | −0.17405 | *** |
| NE—NW | 0.02429 | −0.04308 | 0.09166 | |
| NE—SC | 0.04765 | 0.00053 | 0.09477 | *** |
| NE—NC | 0.36893 | 0.31589 | 0.42197 | *** |
| NW—SE | −0.24533 | −0.30834 | −0.18232 | *** |
| NW—SW | −0.24523 | −0.30860 | −0.18186 | *** |
| NW—NE | −0.02429 | −0.09166 | 0.04308 | |
| NW—SC | 0.02336 | −0.04018 | 0.08690 | |
| NW—NC | 0.34464 | 0.27659 | 0.41269 | *** |
| SC—SE | −0.26869 | −0.30933 | −0.22804 | *** |
| SC—SW | −0.26859 | −0.30980 | −0.22739 | *** |
| SC—NE | −0.04765 | −0.09477 | −0.00053 | *** |
| SC—NW | −0.02336 | −0.08690 | 0.04018 | |
| SC—NC | 0.32128 | 0.27319 | 0.36936 | *** |
| NC—SE | −0.58997 | −0.63735 | −0.54258 | *** |
| NC—SW | −0.58987 | −0.63773 | −0.54201 | *** |
| NC—NE | −0.36893 | −0.42197 | −0.31589 | *** |
| NC—NW | −0.34464 | −0.41269 | −0.27659 | *** |
| NC—NC | −0.32128 | −0.36936 | −0.27319 | *** |

Comparisons that are Significant at the 0.05 Level Are Indicated by ***.

## *3.2. ML Models' Accuracy Evaluation*

The indicator for comparing the accuracy of forecasts of individual models was the average squared error (ASE) indicator. The model with the lowest value of this indicator was evaluated as the most suitable model for prediction.

Table 5 compares the prediction accuracy results of all estimated models, according to which the ASE chooses the model with the smallest average squared error value. The comparison is made using the average squared error of the estimate, which is calculated as the square of the difference between the actual and predicted value of the target variable. The lowest value of the criterion is considered as the best one, because it is the most accurate estimation. The second column shows the accuracy that was achieved when estimating the model (Train). In this case, higher accuracy is usually always achieved in the case of model validation and training. The first column shows the value of the average squared error achieved when predicting the validation sample, and the last column of Table 5 shows the error values achieved when training the model on another data sample as large as the validation sample. These values are slightly higher than in the model estimate. The model that achieved the lowest average square error in PE prediction was Dmine regression. Both models of the neural network were evaluated as the least suitable models for the prediction of PE values.

**Table 5.** Comparison and evaluation of the individual machine learning models ranking from best fit to least suitable. The average squared error (ASE) chooses the model with the smallest average squared error value.

| ML Model | Valid Average Squared Error | Train Average Squared Error | Test Average Squared Error |
|---|---|---|---|
| Dmine Regression | 0.78819 | 0.77826 | 0.78094 |
| Gradient Boosting | 0.79695 | 0.78867 | 0.79537 |
| Decision Tree | 0.84862 | 0.81904 | 0.84500 |
| Ensemble model | 0.93492 | 0.93011 | 0.93512 |
| DM Neural | 1.13691 | 1.14073 | 1.14558 |
| LARS | 1.38220 | 1.37880 | 1.38476 |
| AutoNeural | 1.62204 | 1.61178 | 1.63309 |
| Neural Network | 1.62526 | 1.61762 | 1.63568 |

Figure 5 compares the results of the average predicted values according to the individual models. With the three best models (Dmine regression, gradient boosting, and decision tree), a very similar development of predicted values can be observed. The most significant differences in the predicted values were recorded at higher values; on the contrary, at low predicted values, the resulting predictions of all models were very similar.

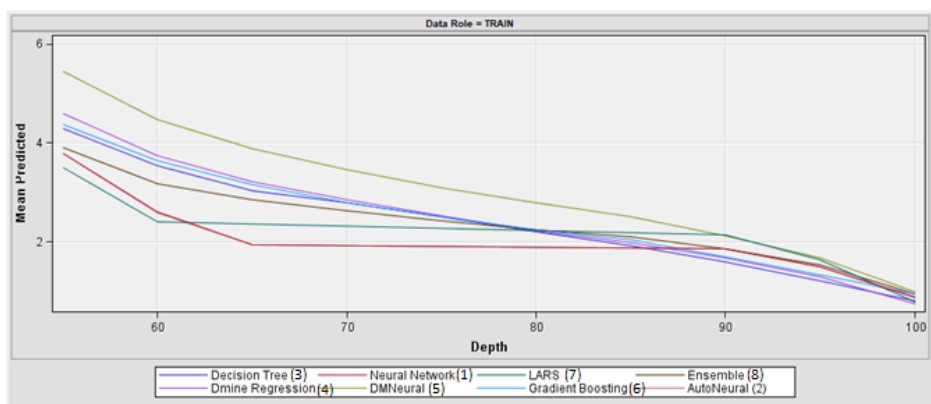

**Figure 5.** Comparison of the individual machine learning models: (1) neural network, (2) autoneural network, (3) decision tree, (4) Dmine regression, (5) DM neural network, (6) gradient boosting, (7) least angle regression, and (8) ensemble model.

Figure 6 indicates the importance of the individual input variables for the PE value according to the most accurate model (Dmine regression). The results show that humidity and temperature have the highest effect on the PE results. All other factors are considered to be far less important. The term OOV16 means that there are variables through which some numeric variables, such as binned variables, are expressed, i.e., categorical variables are created according to numerical variable values, for example. The RH shows a significant difference in the impact of the different expressions.

For comparison, the results according to the second best model are also presented (gradient boosting). The order of the importance of the variables in this case was almost identical to the results obtained with the decision tree. Even in this case, humidity and temperature have the greatest influence (although there is a difference in the importance of the individual variables related to temperature, compared to the first model). In the third case, the elevation above sea level is in third place. Significant variables selected in the Dmine regression model, which was considered superior, are shown in Table 6. The table includes information about the explanatory ability of each variable, the sum of squares,

and its significance expressed by the F-value and *p*-value. Variables denoted as AOV16 represent the categorical expression of quantitative variables. They were used in cases in which this kind of variable achieved better results than the original one. According to the results in Table 6, it can be concluded that the most significant variables influencing pan evaporation were average humidity and minimum temperature. Other variables were also significant, but with much smaller influence on the target variable.

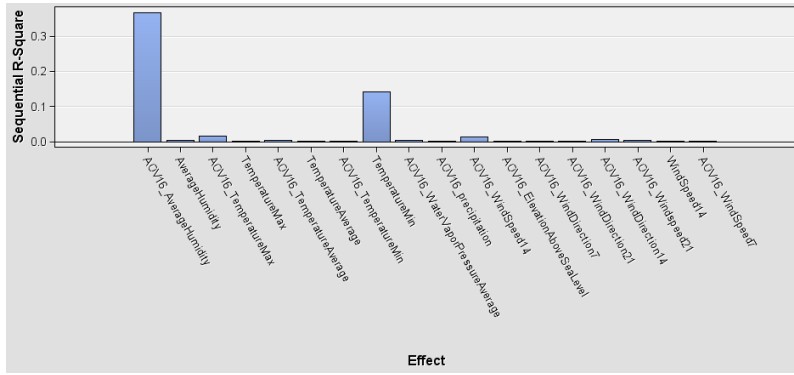

**Figure 6.** Overview of the individual input variables' importance for the PE value assessment according to the most accurate model, Dmine regression.

**Table 6.** The results according to the best model, Dmine regression.

| Effect | DF | R-Square | F Value | *p*-Value | Sum of Squares |
|---|---|---|---|---|---|
| AOV16: Average Humidity | 15 | 0.367893 | 1290.510136 | <0.0001 | 22095 |
| Var: Temperature Min | 1 | 0.142492 | 9679.358776 | <0.0001 | 8557.782572 |
| AOV16: Temperature Max | 10 | 0.015878 | 111.436181 | <0.0001 | 953.574047 |
| AOV16: WindSpeed14 | 15 | 0.012232 | 58.721617 | <0.0001 | 734.603369 |
| AOV16: WindDirection14 | 15 | 0.006584 | 32.049942 | <0.0001 | 395.401311 |
| AOV16: Windspeed21 | 14 | 0.003448 | 18.115361 | <0.0001 | 207.096720 |
| AOV16: Temperature Average | 9 | 0.002904 | 23.875931 | <0.0001 | 174.387981 |
| AOV16: Water Vapor Pressure—Average | 14 | 0.002906 | 15.453458 | <0.0001 | 174.513166 |
| Var: Average Humidity | 1 | 0.002524 | 189.014114 | <0.0001 | 151.605476 |
| AOV16: WindDirection21 | 15 | 0.002285 | 11.461461 | <0.0001 | 137.246991 |
| AOV16: Elevation Above Sea Level | 10 | 0.001970 | 14.880498 | <0.0001 | 118.297418 |
| Var: Temperature Max | 1 | 0.001751 | 132.805014 | <0.0001 | 105.159677 |
| AOV16: Wind Direction7 | 15 | 0.001362 | 6.905653 | <0.0001 | 81.803511 |
| AOV16: Temperature Min | 9 | 0.001215 | 10.295476 | <0.0001 | 72.991107 |
| AOV16: Wind Speed7 | 15 | 0.000957 | 4.873271 | <0.0001 | 57.482001 |
| AOV16: Precipitation | 15 | 0.000854 | 4.355927 | <0.0001 | 51.301760 |
| Var: Wind Speed14 | 1 | 0.000800 | 61.284456 | <0.0001 | 48.030878 |
| Var: Temperature—Average | 1 | 0.000740 | 56.825843 | <0.0001 | 44.461511 |

This study's results are in accordance with other studies, e.g., [39,62–64], which concluded that ML may be a powerful tool for the prediction of actual evapotranspiration when a time series of a few years is available. Starting from the measurements of a sufficient number of climatic parameters, it is possible to obtain forecasting models characterized by very high accuracy and precision.

The final model was selected as the best one, from all common data mining techniques usually used in the modeling of large-scale data. The dataset was, in this case, compiled

from measurements in different places and included a large number of observations. The overall explanatory ability of this model measured by R-squared analysis is slightly smaller than the value presented by other authors; however, this could be explained by a very large number of observations used in the presented analysis, which was unique especially due to the extent of the data (both in time and space). The presented model can be used as the universal model for the prediction of pan evaporation in the Slovak Republic.

### 3.3. Relationship between PE and Variables

Concerning the relationship between PE, temperature and elevation above sea level, the correlation analysis was simplified; only $T_{min}$ and $T_{max}$ were included instead of $T_{aver}$. Table 7 shows that there is a significant yet weak relationship between PE and elevation above sea level. On the other hand, there is a moderately strong relationship between PE and $T_{aver}$, which is a moderate relationship. Comparing $T_{min}$ and $T_{max}$ clearly shows that $T_{max}$ is a slightly more important factor.

**Table 7.** Comparison of the strongest relationship variables and PE on the basis of the Pearson correlation coefficient.

| Pearson Correlation Coefficients | |
| :---: | :---: |
| **Prob > |r| under H0: Rho = 0** | |
| **Number of Observations** | |
| | **PE** |
| $T_{max}$ (°C) | 0.60209 |
| $T_{-max}$ (°C) | <0.0001 |
| | 77,531 |
| $T_{min}$ (°C) | 0.43345 |
| $T_{-min}$ (°C) | <0.0001 |
| | 77,531 |
| $T_{aver}$ | 0.58680 |
| $T_{aver}$ (°C) | <0.0001 |
| | 77,525 |
| **Elevation** | −0.11332 |
| **above** | <0.0001 |
| **sea level (m)** | 77,534 |

### 4. Discussion

The results of this study are built on existing evidence of ML models and statistical techniques as useful frameworks for making predictions of complex climatological indices, such as the hydrological PE, as concluded by numerous authors (see Table 1). Investigating outcomes indicate that the ML models perform well in predicting PE at different climatical regions [1,9,13,16,40–42], which is also consistent with this case study. The results according to [11] showed that the ML models have different accuracies in different climates.

ML models based on multiple regression, such as DR in this study, have the best accuracy of results, which is in line with the hypotheses of the previous studies [11,40,42,45]. However, the results do not fit with the theory according to Zounemat-Kermani (2021) [40], where the kriging model, as well as the support vector regression (SVR), radial basis function neural network (RBFNN), and = Levenberg–Marquardt (MLP-ML) models, performed better compared to the RSM (the modified response surface method) and M5Tree (M5 model tree).

Contrary to this study's hypothesized association, in which AN and NN were evaluated as the least suitable models for the prediction of PE, the study of Sudheer (2002) [50] proved that PE values could be reasonably estimated using temperature data only through the ANN technique. The results of this study also contradict the claims of the research [13], in which the estimation results obtained with the functional link artificial neural network (FLANN) model are compared with those obtained by multi-layer artificial neural net-

works (MLANN) and two empirical methods using the same raw data and corresponding features. In summary, the proposed FLANN models provide improved estimates, and they are able to model the daily evaporation process more accurately compared to MLANN. Out of MLANN and FLANN, the FLANN is less complex and better in terms of accuracy. However, among the three agroclimatic zones, the improvement in the plains zone (Raipur) is more visible, as compared to the plateau (Jagdalpur) and hill (Ambikapur) zones of the study area [13].

The study [13] concludes that appropriate FLANN models can also be selected depending upon the availability of climatic parameters. A comparison according to [40] indicates that the models, without local climatic data, provide more scattered estimates. The studies that used more input climate data variables [9,40,44] have low predicted values; on the contrary, the most significant relationships in the predicted values were recorded in studies with less input climatic data variables [11,40,43]. As confirmed by [11], there is a slight difference between RH and $T_{aver}$, as they were much more effective at modeling PE than the other variables in their study. A different study [42] concluded that besides $T_{aver}$, SR had the best performance, followed by models that used RH and, finally, Us. However, the temperature-based models showed the worst performance in their study. From a study conducted by Al-Mukhtar et al. [1], it is evident that the temperatures ($T_{max}$, $T_{min}$, and $T_{aver}$), RH and Us were all significantly associated at the 0.05 level with the PE, justifying that the PE rates at these stations can be nicely modelled by these elements. The results of this study are consistent with the above-mentioned studies and proved the strong relationship between PE and $T_{aver}$.

The results of this study match with the study of Majhi and Naidu (2021) [13], where they stated that the evaporation rate increases with an increase in the radiation and temperature of the evaporating zone. Similarly, wind speed also helps to remove the water from water bodies to some extent and triggers the evaporation process. Bright sunshine and low humidity and atmospheric pressure are the other climatic variables that significantly contribute to the evaporation process [13]. The study used the long-term climatic data on $T_{max}$ (°C), $T_{min}$ (°C), and RH for morning and afternoon hours (RH1 and RH11 (%)), and Up, SR, and PE were collected from the certified observatories located in selected stations in India [13].

The different studies used different numbers of meteorological stations for their calculations, from one [44,45] or two [9,16,40] up to eight [11,42]. However, in this case study, 35 meteorological stations were used to improve the calculation accuracy, which provides new insight and is an advantage of this study.

In a study from the same country as this case study (Košice, Slovak Republic) [35], the soft computing techniques for estimating daily PE were investigated, using daily SR, RH, T, and Us as the meteorological variables for modeling. These model results show that the ML model performs better than another study that used soft computing techniques [35]; this was the main motivation of this study. However, in this case study, the applicability of the newly explored ML models was investigated, with the best accuracy obtained by DR. For future research, it can be compared with the best performing models of ML from Table 1. The macro-regions with the greatest change in PE will be used for further research applying PE with other variables (SE and SW regions of the Slovak Republic) with the uniform climatic characteristics.

In general terms, no one ML algorithm is the best for all problems [50]. The performance of different machine learning algorithms strongly depends on the size and structure of the available data. As concluded in the study by Granata (2019) [50], this study also infers that the individual ML models have been chosen because they usually achieve high performance and they are very good at learning complex, highly non-linear relationships. However, due to the large dispersion of climatic, geographical, and local conditions, it is necessary to test individual ML models independently for each site.

## 5. Conclusions

This study explored the abilities of eight different ML models—NN, AN, DT, DR, DM NN, GB, LARS, and EM—in modeling PE, utilizing the meteorological dataset combinations of the variables $T_{min}$, $T_{max}$, $T_{aver}$, RH, and Us with wind speed in the directions at 7, 14, and 21 degrees; prevailing wind directions at 7, 14, and 21 degrees; PE; elevation above sea level; and geographical location. The climatic datasets obtained from 35 stations in different geographical locations were used as inputs for training and testing. The datasets were taken from eight regions in the Slovak Republic during the vegetation periods of the years from 2010 to 2020.

The eight regions were classified into six macro-regions—NW, SW, NC, SC, NE, and SE—according to the agroclimatic zones of the Slovak Republic. The results of this study could be applied practically in the field of regional PE estimation. According to results of the performed analysis, the best method for the modeling of pan evaporation was Dmine regression. Both neural network models were the worst performing models from all applied methods. This suggests that the character of the modelled relationships could be explained better by other methods, despite the ability of neural networks to explain complex relationships. On the other hand, Dmine regression allows us to use a standard modeling approach and offers better insight into the modelled relationships, which is not the case of the "black box" neural network.

From this study, we conclude the following:

(a) The lowest PE values were recorded in the NC area, and the highest were recorded in the SE and SW regions. The largest PE change over the observed period (expressed by using the variation range) occurred in the SE region, followed by SC and SW. The smallest PE change was in the NC and NW regions.

(b) The best accuracy of the ML models was obtained by DR (TASE = 0.78819), followed by GB (TASE = 0.77826) and DT (TASE = 0.78094), though it is possible to see very similar results of the predicted values. Both neural network models, AN and NN, were evaluated as the least suitable models for the prediction of PE.

(c) There is a significant but weak relationship between PE and elevation above sea level. However, there is a moderately strong relationship between PE and $T_{aver}$. A comparison between $T_{min}$ and $T_{max}$ shows that $T_{max}$ is a slightly more important factor.

Because ET is one of the main parts of the hydrology cycle, it is crucial for estimating PE, especially when there are limited data available. Based on the results, the use of the best three evaluated ML models is recommended (DR, GB, and DT) in the region with the largest PE change and with uniform climatic characteristics. The previous studies that used more input climate data variables have low predicted values; on the contrary, the most significant relationships in the predicted values were recorded in studies with less input climatic data variables.

The results of this study can be used in water resources management to make sustainable irrigation plans, design sustainable water supply systems, or carry out sustainable reservoir management.

**Author Contributions:** Conceptualization: Ľ.J., B.N. and J.Č.; methodology, Ľ.J., B.N., J.Č. and J.P.; investigation, Ľ.J., B.N., J.Č., J.P. and B.C.; resources, B.N., J.Č. and J.P.; data curation: J.P., B.C. and V.K.; writing—original draft preparation: B.N. and J.P.; writing—review and editing: Ľ.J., B.N., J.Č., J.P. and B.C.; supervision: Ľ.J.; project administration, B.N. and J.Č.; funding acquisition B.N. and J.Č. All authors have read and agreed to the published version of the manuscript.

**Funding:** This publication was funded by the Cultural and Educational Grant Agency KEGA Project no. 29SPU-4/2020: Interactive learning as a tool for analysis and solutions design for the water modeling training in the landscape, and no. 026SPU-4/2020: Climate change and its impact on temperature conditions in Slovakia. At the same time, it is the result of the project implementation no. ITMS2014+313011W580: Scientific support of climate change adaptation in agriculture and mitigation of soil degradation supported by the Integrated Infrastructure Operational Programme funded by the ERDF.

**Institutional Review Board Statement:** This study used secondary data and ethical approval was not required.

**Informed Consent Statement:** Not applicable.

**Data Availability Statement:** The data generated during and/or analyzed during the current study are available from the corresponding author on reasonable request.

**Conflicts of Interest:** The authors declare no conflict of interest.

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
