# Peer review of "Machine Learning for Pan Evaporation Modeling in Different Agroclimatic Zones of the Slovak Republic (Macro-Regions)"

_sustainability, doi:10.3390/su14063475_

Round 1
Reviewer 1 Report
Thank you very much for the opportunity to read and analyze the work. This is an interesting study, conducted with a huge amount of data that could provide valuable results in the future.
However, this work cannot be accepted in a journal of such high level as Sustainability since the article lacks an approach that tries to contextualize it in a previous current of thought. Without this theoretical development, without contextualizing the results in the previous currents of thought, their value is scarce.
An exhaustive theoretical development on this subject is necessary, which contextualizes the empirical study presented here, and whose results corroborate or contradict previous studies on the subject. I believe that what has been said above is corroborated by the absence of a discussion section in the paper, nor an indication of the usefulness of these results for the different agents involved.
I think the work should be completely reworked and focused on a previous school of thought. I encourage researchers to continue working on this line.
Best regards
Author Response
Dear reviewer,
I would like to thank you for your valuable comments and recommendations. I really appreciate your effort to improve the quality of our manuscript.
All your suggestions were incorporated. According to your remarks the manuscript was modified. In its actual version the research issues are better presented as well as the current state of research in this field. For the manuscript conceptualization it was added information about the used approaches and manuscript results, those are described in detail.
I look forward to hearing from you in case of any further comments you may have.
Sincerely,
BN
Reviewer 2 Report
- What is the benefit of using ANN Technique?
- Its looked better if, author compare their results to others work.
- Future scope should be added.
- Even though the authors briefly mentioned, it would be better to provide industrial or engineering examples where the proposed approach can be used.
- The introduction section does not highlight your contribution. Please mention your contributions/novelty clearly with bullet points in introduction section.
- The limitations of the proposed model should be added conclusion section.
Author Response
Please see our reponse in the attachemnet.

Reviewer 3 Report
The authors are thanked for the work carried out and the effort put into it, although at the discretion of the writer of these lines, the authors should emphasize the methodology used, the preparation of the data and the results obtained.
Therefore, a better methodological description is recommended, perhaps a flow chart could help, as well as, a detailed description of the algorithms developed and the used tools, which would allow to properly observe the magnitude of the developed work .
It would also be advisable to include more details in the zoning criteria and the treatment of the information, as well as the spatial data were integrated in the analyzes carried out.
The authors are again thanked and encouraged to move on.
Author Response
Please find our response in the attachment.

Round 2
Reviewer 1 Report
Thank you for giving me the opportunity to read this interesting work again. The authors have made an effort to contextualize their study within the related literature, but have not linked this prior literature to the results of the article. Do the results of your study confirm, contradict, complement other previous studies, other previous theoretical approaches? All of this should be included in a specific Discussion section that does not exist but is necessary in studies of this nature.
Author Response
Dear reviewer,
I would like to thank you for spending your time again to review our manuscript as well as for giving us opportunity to revise it and to improve its quality. Your valuable comments we have incorporated into the text of the manuscript.
In the abstract it was added information that the global climate change has caused changes that have initiated many evapotranspiration calculation methods to fail. Therefore, to verify the suitability of the calculation methods, we used a machine learning model to determine the most appropriate relationships for the evaporation calculation in global climate change conditions.
As you recommended, we added the information about the related literature resources linked to our manuscript. Please see Table 1 and newly added text in the page 2, 3, 4, 17 and 18. We have fixed small errors and changes in the text and added a description in Fig. 5, which is now complete. The changes are marked and tracked in the text.
Please be so kind to read, how we have integrated the changes you have proposed into the text. Thank you.
Sincerely Yours
Beáta Novotná
Reviewer 3 Report
The authors have improved the presentation of the paper, allowing an adequate review of it.Although it is an interesting article, the degree of advancement in science is small.
But the effort of the authors to improve the presentation of the paper is recognized.
Author Response
Dear reviewer,
I would like to thank you for your valuable comments as well as for taking your time to assess our manuscript again.
Would you be so kind to reconsider your opinion, please? In the manuscript are presented newly developed machine learning models, those were not used for the evaporation modelling yet. This study is unique in Slovak Republic.
In the abstract it was added information that the global climate change has caused changes that have initiated many evapotranspiration calculation methods to fail. Therefore, to verify the suitability of the calculation methods, we used a machine learning model to determine the most appropriate relationships for the calculation of evaporation in the global climate change conditions.
We added information about the related literature resources linked to our manuscript. Please see Table 1 and newly added text in the page 2, 3, 4, 17 and 18. We have fixed the small errors and changed Table 5, which is now complete. The changes are marked and tracked in the text.
We have performed all the necessary changes to increase the manuscript scientific value and to make it more attractive for the scientific community.
Sincerely Yours
Beáta Novotná
Round 3
Reviewer 1 Report
The author has tried to develop the Discussion section but for this he/she/they must carry out a greater and deeper analysis of the previous literature on the line of research on which the work is based. Without this deep theoretical development, the importance and contribution of this work cannot be justified. Comparing the results of the study with a single previous work is, IMHO, not enough at all.
Author Response
Dear reviewer,
I would like to thank you for giving us opportunity to improve considerably our manuscript. The section Discussion was better developed and extended as you suggested. I am pretty sure that your valuable comments and advice increased the quality of the manuscript and made it more attractive to the reader. Finally, by it has been expanded its scientific value.
In case of any other suggestions, we are ready to incorporate them into the manuscript text.
Sincerely Yours
Beáta Novotná
Round 4
Reviewer 1 Report
The author has adequately followed the recommendations and has made an effort to search for references related to the theme that is developed. However, he has not carried out an exhaustive analysis of the state of the art, since the incorporated references are very recent (most are from the year 2021). It is necessary to incorporate references obtained from a more complete analysis, which should also be included in the Discussion section, comparing them with the results obtained in this work. Thank you for the opportunity to read the work again.
Author Response
Dear reviewer,
I would like to thank you for your answer as well as for your valuable comments concerning to our manuscript. They gradually increase its quality.
I would like to inform you that we added information about the ML history in the introduction part (lines 114-119) as well as the justification, why chosen ML models have been used in the discussion part (lines 523-529) in enclosed Word document – version 4th. However, given problematic of the ML application for the evapotranspiration estimation has not been used in long term. Therefore, all the available literature resources have been used as they are from the recent period of the years from 2015 to 2021. As it is concluded, in general terms, no one ML algorithm is the best for all problems. Due to the large dispersion of climatic, geographical, and local conditions, it is necessary to test individual ML models independently for each specific site.
I wish you a lot of success in your future work.
Sincerely Yours
Beáta Novotná